# Covasim: An agent-based model of COVID-19 dynamics and interventions

**Cliff C. Kerr**[1]*, **Robyn M. Stuart**[2,3°], **Dina Mistry**[1°], **Romesh G. Abeysuriya**[3], **Katherine Rosenfeld**[1], **Gregory R. Hart**[1], **Rafael C. Núñez**[1], **Jamie A. Cohen**[1], **Prashanth Selvaraj**[1], **Brittany Hagedorn**[1], **Lauren George**[1], **Michał Jastrzębski**[4], **Amanda S. Izzo**[1], **Greer Fowler**[1], **Anna Palmer**[3], **Dominic Delport**[3], **Nick Scott**[3], **Sherrie L. Kelly**[3], **Caroline S. Bennette**[1], **Bradley G. Wagner**[1], **Stewart T. Chang**[1], **Assaf P. Oron**[1], **Edward A. Wenger**[1], **Jasmina Panovska-Griffiths**[5,6], **Michael Famulare**[1], **Daniel J. Klein**[1]

**1** Institute for Disease Modeling, Global Health Division, Bill & Melinda Gates Foundation, Seattle, Washington, United States of America, **2** Department of Mathematical Sciences, University of Copenhagen, Copenhagen, Denmark, **3** Burnet Institute, Melbourne, Victoria, Australia, **4** GitHub, Inc., San Francisco, California, United States of America, **5** Big Data Institute, University of Oxford, Oxford, United Kingdom, **6** Wolfson Centre for Mathematical Biology and The Queen's College, University of Oxford, Oxford, United Kingdom

☯ These authors contributed equally to this work.

* info@covasim.org

**Data Availability Statement:** The Covasim model code and documentation is fully open source and available via GitHub: https://github.com/institutefordiseasemodeling/covasim.

**Funding:** The author(s) received no specific funding for this work.

## Abstract

The COVID-19 pandemic has created an urgent need for models that can project epidemic trends, explore intervention scenarios, and estimate resource needs. Here we describe the methodology of Covasim (COVID-19 Agent-based Simulator), an open-source model developed to help address these questions. Covasim includes country-specific demographic information on age structure and population size; realistic transmission networks in different social layers, including households, schools, workplaces, long-term care facilities, and communities; age-specific disease outcomes; and intrahost viral dynamics, including viral-load-based transmissibility. Covasim also supports an extensive set of interventions, including non-pharmaceutical interventions, such as physical distancing and protective equipment; pharmaceutical interventions, including vaccination; and testing interventions, such as symptomatic and asymptomatic testing, isolation, contact tracing, and quarantine. These interventions can incorporate the effects of delays, loss-to-follow-up, micro-targeting, and other factors. Implemented in pure Python, Covasim has been designed with equal emphasis on performance, ease of use, and flexibility: realistic and highly customized scenarios can be run on a standard laptop in under a minute. In collaboration with local health agencies and policymakers, Covasim has already been applied to examine epidemic dynamics and inform policy decisions in more than a dozen countries in Africa, Asia-Pacific, Europe, and North America.

**Competing interests:** The authors have declared that no competing interests exist.

## Author summary

Mathematical models have played an important role in helping countries around the world decide how to best tackle the COVID-19 pandemic. In this paper, we describe a COVID-19 model, called Covasim (COVID-19 Agent-based Simulator), that we developed to help answer these questions. Covasim can be tailored to the local context by using detailed data on the population (such as the population age distribution and number of contacts between people) and the epidemic (such as diagnosed cases and reported deaths). While Covasim can be used to explore theoretical research questions or to make projections, its main purpose is to evaluate the effect of different interventions on the epidemic. These interventions include physical interventions (mobility restrictions and masks), diagnostic interventions (testing, contact tracing, and quarantine), and pharmaceutical interventions (vaccination). Covasim is open-source, written in Python, and comes with extensive documentation, tutorials, and a webapp to ensure it can be used as easily and broadly as possible. In partnership with local stakeholders, Covasim has been used to answer policy and research questions in more than a dozen countries, including India, the United States, Vietnam, and Australia.

## 1 Introduction

More than a year after COVID-19 was first identified, governments continue to be faced with an urgent need to understand the rapidly evolving pandemic landscape and translate it into policy. Since the onset of the pandemic, mathematical modeling has been at the heart of informing this decision-making. Numerous statistical models and data visualization tools have been developed over the last year in an attempt to meet this demand, with varying purposes, structures, and levels of detail and complexity; for example, despite their limitations [1], data dashboards have proven crucial for understanding the current state of the epidemic on both global and local scales [2,3]. However, more detailed models are needed to evaluate scenarios based on complex intervention strategies. These strategies are important to evaluate in order to understand the epidemiological impact of reopening schools, businesses, and society.

Models for examining COVID-19 transmission and control measures can be broadly divided into two main types: compartmental models and agent-based models (also called individual-based or microsimulation models), with the former generally being simpler and faster, while the latter are generally more complex, detailed, and computationally expensive. Numerous compartmental models have been developed or repurposed for COVID-19: Walker et al. [4] used an age-structured stochastic "susceptible, exposed, infectious, recovered" (SEIR) model to determine the global impact of COVID-19 and the effects of various social distancing interventions; Read et al. [5] developed an SEIR model to estimate the basic reproduction number in Wuhan; Keeling et al. [6] used one to look at the efficacy of contact tracing as a containment measure; and Dehning et al. [7] used an SIR model to quantify the impact of intervention measures in Germany. In models such as those by Giordano et al. [8] and Zhao and Chen [9], compartments are further divided to provide more nuance in simulating progression through different disease states, and have been deployed to study the effects of various population-wide interventions such as social distancing and testing on COVID-19 transmission.

For microsimulation models, several agent-based influenza pandemic models have been repurposed to simulate the spread of COVID-19 transmission and the impact of social distancing measures in the United Kingdom [10], Australia [11], Singapore [12], and the United States [13]. Additionally, new agent-based models have been developed to evaluate the impact

of social distancing and contact tracing [14–18] and superspreading [19]. Features of these models include accounting for the number of household and non-household contacts [13,15,16]; the age and clustering of contacts within households [13,14,16]; and the microstructure in schools and workplace settings informed by census and time-use data [14]. Branching process models have also been used to investigate the impact of non-pharmaceutical intervention strategies [20,21] and the proportion of unobserved infections [22].

In developing Covasim, our aim was to produce a tool that would be capable of informing real-world policy decisions for a variety of national and subnational settings. We wanted to capture the benefits of agent-based modeling (in particular, the ability of such models to simulate the kinds of microscale policies being used to respond to the COVID-19 pandemic), whilst making use of recent advances in software tools and computational methods to minimize the complexity and computation time typically associated with such models. In this regard, Covasim is most similar to the OpenABM-Covid19 model [23,24], which has also been developed as a high-performance, user-friendly, general-purpose COVID model.

Overall, the design principle we followed with Covasim was to make common usage patterns as simple as possible, while still giving the user the ability to customize virtually all aspects of the simulation. For example, Covasim comes pre-loaded with demographic data for each country (Section 2.4), but users can also define custom populations and contact networks down to the level of a single city [25] or even university [26]. Common COVID-19 interventions are built into Covasim (Section 2.5), and custom interventions of arbitrary complexity can also be defined. In addition, Covasim's high performance for an agent-based model, achieved via dynamic rescaling (Section 2.6.2) and array-based computations (Section 2.7.1), means that most analyses can be run on a standard laptop, removing the need to use a high-performance computing cluster except for large parameter sweeps or model calibrations (Section 2.6.8). These design choices are intended to allow users to start running simple Covasim analyses quickly, while providing flexibility later if more detailed data become available or if the modeling questions become more nuanced.

To date, Covasim has been used by researchers and public health officials in over a dozen countries. Covasim has been used to inform policy decisions in the United States [25,27], Vietnam [28], the United Kingdom [29], and Australia [30]. It has also been used for research studies in these locations [31–34], and well as other countries including India, Russia, Kenya, and South Africa. This paper describes the methodology underlying Covasim, and provides several examples illustrating its use, including an application to Seattle where Covasim scenarios were used to inform a rapid policy decision, with subsequent validation of these findings by real-world data.

## 2 Design and implementation

### 2.1 Design and implementation

Covasim simulates the state of individual people, known as agents, over a number of discrete time steps. Conceptually, the model is largely focused on a single type of calculation: the probability that a given agent on a given time step will change from one state to another, such as from susceptible to infected, or from critically ill to dead. Once these probabilities have been calculated, a pseudorandom number generator with a user-specified seed is used to determine whether the transition actually takes place for a given model run.

The logical flow of a single Covasim run is as follows. First, the simulation object is created, then the parameters are loaded and validated for internal consistency, and any specified data files are loaded (described in Section 2.6.1). Second, a population of agents is created, including age, sex, and comorbidities for each agent, drawing from location-specific data

distributions where available; then, agents are then connected into contact networks based on their age and other statistical properties (Section 2.4). Next, the integration loop begins. On each timestep (which corresponds to a single day by default), the order of operations is: dynamic rescaling (Section 2.6.2); applying health system constraints (Section 2.6.3); updating the state of each agent, including disease progression (Section 2.2); importation events (Section 2.6.4); applying interventions (Section 2.5); calculating disease transmission events across each infectious agent's contact network (Section 2.3); collating outputs into results arrays (Section 2.6.5); and applying analyzers (Section 2.6.7). The following sections describe each step in more detail.

## 2.2 Disease progression

In Covasim, each individual is characterized as either susceptible, exposed (i.e., infected but not yet infectious), infectious, recovered, or dead, with infectious individuals additionally categorized according to their symptoms: asymptomatic, presymptomatic, mild, severe, or critical. A schematic diagram of the model structure is shown in Fig 1.

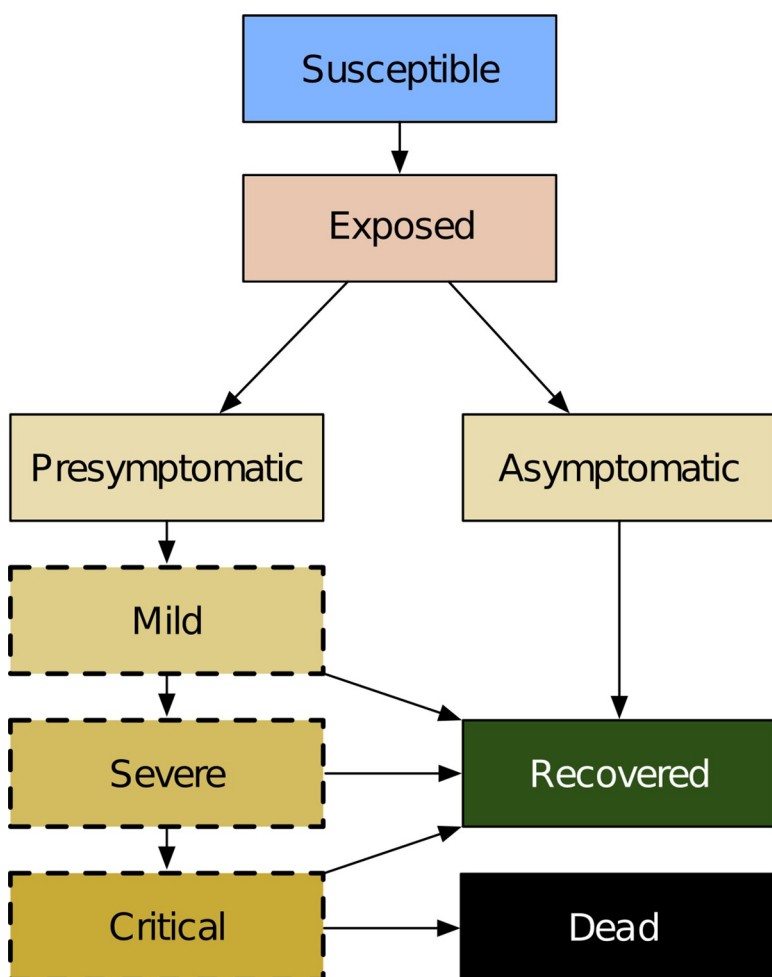

**Fig 1. Covasim model structure, including infection (exposure), disease progression, and final outcomes.** Yellow shading indicates that an individual is infectious and can transmit the disease to other susceptible agents. States with a dashed border are considered symptomatic with respect to symptomatic versus asymptomatic testing.

**Table 1. Default duration parameters, in days, used in the Covasim model.**

| Parameter | Description | Distribution (mean, std) | Source |
|---|---|---|---|
| $\tau_{inf}$ | Exposed-to-infectious: Length of time after exposure before an individual is infectious (i.e., has begun viral shedding) | $\tau_{inf} \sim$ lognormal (4.5, 1.5) | From Lauer et al. [37]; additional sources Du et al., Nishiura et al., and Pung et al. [38–40]. |
| $\tau_{sym}$ | Infectious-to-symptomatic: Length of time after viral shedding has begun before an individual has symptoms | $\tau_{sym} \sim$ lognormal (1.1, 0.9) | Linton et al. [41] report the incubation period as 5.6 days (95% CI: 5.0–6.3 days). Using the period of exposure before becoming infectious, we infer the period of viral shedding before symptomatic. However, other studies have estimated longer periods, e.g. [42]. |
| $\tau_{sev}$ | Symptomatic-to-severe: Length of time after symptoms have started that symptoms have become severe (i.e., requiring hospitalization) | $\tau_{sev} \sim$ lognormal (6.6, 4.9) | Linton et al. [41], Wang et al. [43] |
| $\tau_{cri}$ | Severe-to-critical: Length of time after severe symptoms have started that symptoms have become critical (i.e., requiring ICU) | $\tau_{cri} \sim$ lognormal (1.5, 2.0) | Chen et al. [44], Wang et al. [43] |
| $\tau_{dea}$ | Critical-to-death: Time from onset of critical symptoms to death | $\tau_{dea} \sim$ lognormal (10.7, 4.8) | Verity et al. [45] |
| $\tau_{ra}$ | Time from infectiousness onset to recovery for asymptomatic cases | $\tau_{ra} \sim$ lognormal (8.0, 2.0) | Wölfel et al. [46] |
| $\tau_{rm}$ | Time from symptom onset to recovery for mild cases | $\tau_{rm} \sim$ lognormal (8.0, 2.0) | Wölfel et al. [46] |
| $\tau_{rs}$ | Time from onset of severe symptoms to recovery for severe cases | $\tau_{rs} \sim$ lognormal (18.1, 6.3) | Verity et al. [45] |
| $\tau_{rc}$ | Time from onset of critical symptoms to recovery for critical cases | $\tau_{rc} \sim$ lognormal (18.1, 6.3) | Verity et al. [45] |

The length of time after exposure before an individual becomes infectious is set by default to be a log-normal distribution with a mean of 4.6 days, which is within the range of values reported across the literature (Table 1). The length of time between the start of viral shedding and symptom onset is assumed to follow a log-normal distribution with a mean of 1.1 days (Table 1). Exposed individuals may develop symptoms or may remain asymptomatic. Individuals with symptoms are disaggregated into either mild, severe, or critical cases, with the probability of developing a more acute case increasing with age (Table 2). Covasim can also model the effect of comorbidities, which act by modifying an individual's probability of developing severe symptoms (and hence critical symptoms and death). By default, comorbidity multipliers are set to 1 since they are already factored into the marginal age-dependent disease progression rates.

**Table 2. Age-linked disease susceptibility, progression, and mortality probabilities.** Key: $r_{sus}$: relative susceptibility to infection; $p_{sym}$: probability of developing symptoms; $p_{sev}$: probability of developing severe symptoms (i.e., sufficient to justify hospitalization); $p_{cri}$: probability of developing into a critical case (i.e., sufficient to require ICU); $p_{dea}$: probability of death (i.e., infection fatality ratio). Relative susceptibility values are derived from odds ratios presented in Zhang et al. [47]. Mortality rates are based on O'Driscoll et al. [48] for ages <90 and Brazeau et al. [49] for ages >90. All other values are derived from Verity et al. [45] and Ferguson et al. [50], which did not differentiate 80–89 and 90+. Values were validated from model fits to data on numbers of cases, numbers of people hospitalized and in intensive care, and numbers of deaths from Washington and Oregon states. Note that "overall" values depend on the age structure of the population being modeled. For a population like the USA or UK, the symptomatic proportion is roughly 70%, while for populations skewed towards younger ages, this proportion is lower. Similarly, overall mortality rates are estimated to vary from 0.2% in Kenya to 0.9% in the USA and 1.4% in Italy.

| | 0–9 | 10–19 | 20–29 | 30–39 | 40–49 | 50–59 | 60–69 | 70–79 | 80–89 | 90+ | Overall |
|---|---|---|---|---|---|---|---|---|---|---|---|
| $r_{sus}$ | 0.34 | 0.67 | 1.00 | 1.00 | 1.00 | 1.00 | 1.00 | 1.24 | 1.47 | 1.47 | 1.00 |
| $p_{sym}$ | 0.50 | 0.55 | 0.60 | 0.65 | 0.70 | 0.75 | 0.80 | 0.85 | 0.90 | 0.90 | 0.5–0.75 |
| $p_{sev}$ | 0.00050 | 0.00165 | 0.00720 | 0.02080 | 0.03430 | 0.07650 | 0.13280 | 0.20655 | 0.24570 | 0.24570 | 0.1–0.2 |
| $p_{cri}$ | 0.00003 | 0.00008 | 0.00036 | 0.00104 | 0.00216 | 0.00933 | 0.03639 | 0.08923 | 0.17420 | 0.17420 | 0.05–0.1 |
| $p_{dea}$ | 0.00002 | 0.00002 | 0.00010 | 0.00032 | 0.00098 | 0.00265 | 0.00766 | 0.02439 | 0.08292 | 0.16190 | 0.002–0.015 |

Estimates of the duration of COVID-19 symptoms and the length of time that viral shedding occurs are highly variable, but durations are generally reported to increase according to acuity [35,36]. We reflect this in our model with different recovery times for asymptomatic individuals, those with mild symptoms, and those with severe symptoms, as summarized in Table 1. All non-critical cases are assumed to recover, while critical cases either recover or die, with the probability of death increasing with age (Table 2).

## 2.3 Transmission and within-host viral dynamics

Whenever a susceptible individual comes into contact with an infectious individual on a given day, transmission of the virus occurs with probability $\beta$. For a well-mixed population where each individual has an average of 20 contacts per day, a value of $\beta = 0.016$ corresponds to a doubling time of roughly 4–6 days and an $R_0$ of approximately 2.2–2.7, with the exact value depending on the population size, age structure, and other factors. The value of $\beta = 0.016$ that is currently used as the default in Covasim was based on calibrations to data from Washington and Oregon states. However, this default value is too low for high-transmission contexts such as New York City or Lombardy [51], and may be too high for low-transmission contexts such as India's first wave [52]. Hence, this parameter must be calibrated by the user to match local epidemic data, as described in Section 2.6.8.

If realistic network structure (i.e., households, schools, workplaces, and community contacts) is included, the value of $\beta$ depends on the contact type. Default transmission probabilities are roughly 0.050 per contact per day for households, 0.010 for workplaces and schools, and 0.005 for the community. These values correspond to relative weightings of 10:2:2:1, chosen (a) for consistency with both time-use surveys [53] and studies of infections with known contact types [47], and (b) to have a weighted mean close to the default $\beta$ value of 0.016 for a well-mixed population (i.e., if different network layers are not used). When combined with the default number of contacts in each layer, age-based susceptibility, and other factors, for a typical (unmitigated) transmission scenario, the proportions of transmission events that occur in each contact layer in the absence of interventions are approximately 30% via households, 25% via workplaces, 15% via schools, and 30% via the community. The value of $\beta$ can also be modified by interventions, such as physical distancing, as described below.

In addition to allowing individuals to differ in terms of disease severity and time spent in each disease state, we allow individual infectiousness to vary between people and over time. We use individual viral load to model these differences in infectivity. Several studies have found that viral load is highest around or slightly before symptom onset, and then falls monotonically [54–58]. As a simple approximation to this viral time course, we model two stages of viral load: an early high stage followed by a longer low stage. By default, we set the viral load of the high stage to be twice as high as the low stage and to last for either 30% of the infectious duration or 4 days, whichever is shorter. The default viral load for each agent is drawn from a negative binomial distribution with mean 1.0 and shape parameter 0.45, which was the value most consistent with both international estimates [59,60] and fits to data in Washington state and Oregon. The daily viral load is used to adjust the per-contact transmission probability ($\beta$) for an agent on a given day. Viral loads for a representative sample of individuals given default parameter values are shown in Fig 2. The proportion of transmissions by asymptomatic, presymptomatic, and symptomatic individuals varies by context; estimated proportions for Seattle are shown in [25].

Evidence is mixed as to whether transmissibility is lower if the infectious individual does not have symptoms [55]. We take a default assumption that it is not, but include a parameter that can be modified as needed depending on the modeling application or context, noting that some studies have used much lower rates of infectiousness for asymptomatic individuals [61].

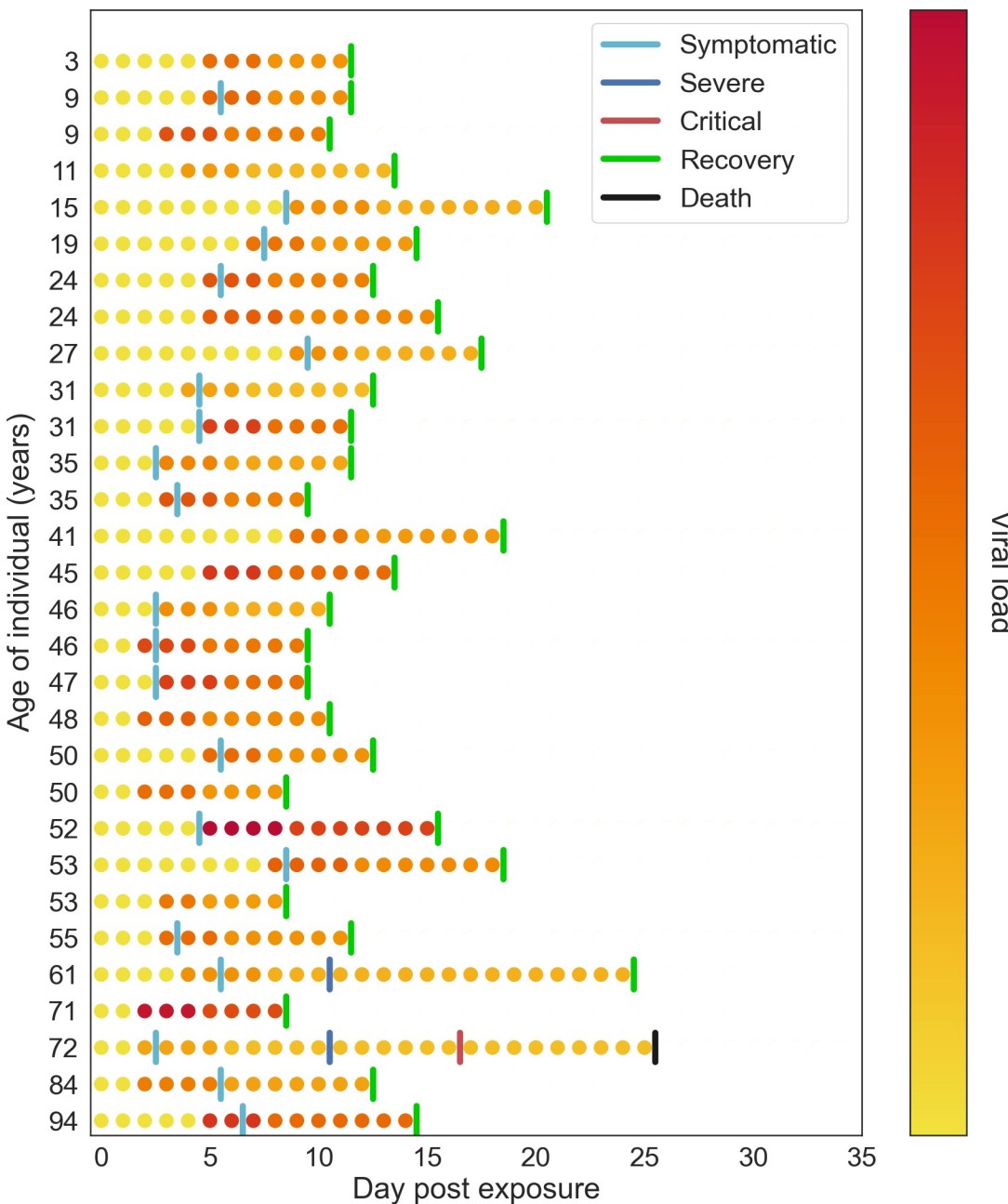

**Fig 2. Example of within-host viral load dynamics in Covasim.** Each row shows a different agent in the model. Color indicates viral load, which typically peaks the day before or the day of symptom onset, before declining slowly.

## 2.4 Contact network models

Covasim is capable of generating and using three alternative types of contact networks: random networks, SynthPops networks, and hybrid networks. Each of these may be useful in different settings, and in addition users have the option of defining their own networks. Covasim's default contact networks are shown schematically in Fig 3; different options for construction these networks are provided in the following sections. To facilitate easy adaptation to different contexts, Covasim comes pre-loaded with data on country age distributions and household sizes as reported by the UN Population Division 2019 (population.un.org).

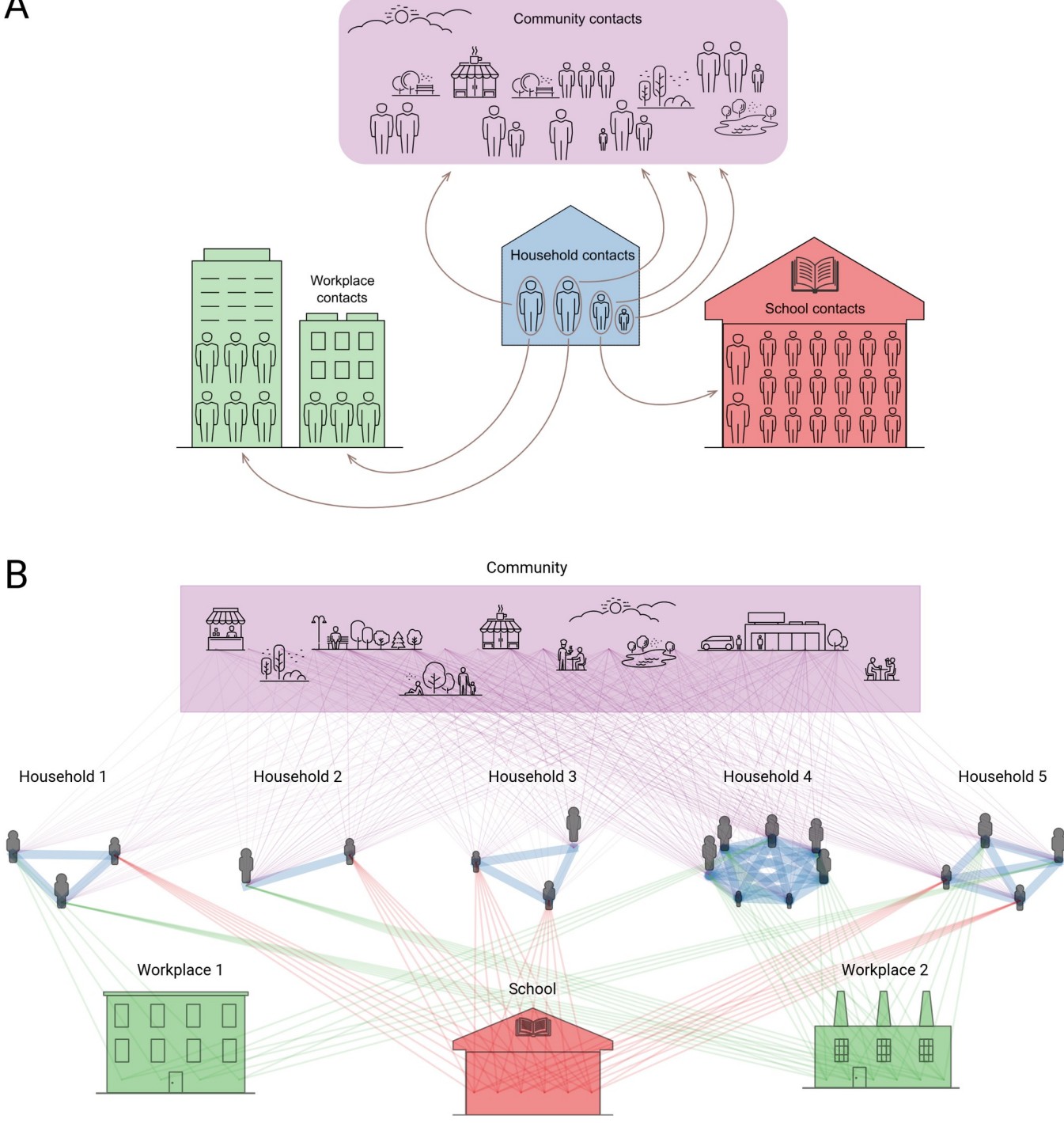

**Fig 3. Illustration of contact networks with multiple layers in Covasim.** (A) In reality, individual people move between household, school, workplace, and community contact layers during the day. (B) In the model (shown here with a population of 20 people, with age structure and household sizes based on Malawi data), these dynamic contacts are approximated as static average daily contacts between layers. Individuals have different numbers of connections (lines) and connection weights (line widths; default relative weights shown) for each layer.

**2.4.1 Random networks.** Covasim generates random networks by assuming that each person in the modeled population can come into contact with anyone else in the population. Each person is assigned a number of daily contacts, which is drawn from a Poisson distribution whose mean value can be specified by the user depending on the modeling context (with a default value of 20). The user can also decide whether these contacts should remain the same throughout the simulation, or whether they should be sampled randomly from the population each day.

**2.4.2 SynthPops networks.** Covasim is integrated with SynthPops, an open-source data-driven model capable of generating realistic synthetic contact networks for populations; further information, including documentation and source code, is available from synthpops.org. Briefly, the method draws upon previously published models and empirical studies to infer high-resolution age-specific contact patterns in key settings (e.g., households, schools, workplaces, and the general community) relevant to the transmission of infectious diseases [62–64]. Census or survey data such as those from Demographic and Health Surveys [65,66] are used by SynthPops to inform demographic characteristics (e.g., age, household size, school enrollment, and employment rates). Age-specific contact matrices, such as those in [62,67–69], are then used to generate individuals and their expected contacts in a multilayer network framework. By default, SynthPops generates household, school, and work contact networks; community connections are generated using the random approach described above, and long-term care facilities can be included if data are available. An example synthetic network as generated by SynthPops is shown in Fig 4.

**2.4.2.1 *Households*.** SynthPops generate individuals within households using data on the distribution of ages, household sizes, and the age of reference individuals per household for a given population. The algorithm first generates household sizes from the household size distribution, and then assigns a reference individual (for example, the head of the household) with their age sampled conditional on the household size. To construct the other household members, location-specific household age mixing contact matrices and the population age distribution are used to infer the likely ages of household contacts for the reference person. Each column *c* of the contact matrix is treated as an age distribution of the household contacts for a person in the age group *c*. The ages of other household members are then sampled conditional on the age of the reference person for the household.

**2.4.2.2 *Schools*.** A similar approach is used to construct schools. School enrollment data, available from census studies or survey data can be used to inform enrollment rates by age, school sizes, and student-teacher ratios. The SynthPops algorithm first chooses a reference student for the school conditional on enrollment rates to infer the school type, and then uses the age mixing contact matrix in the school setting to infer the likely ages of the other students in the school. Students are drawn from an ordered list of households, such that they reproduce an approximation of the neighborhood dynamics of children attending area schools together. Teachers and other non-teaching staff (e.g., administrative or cleaning staff) are drawn from the adult population comprising the labor force and assigned to schools as needed, reflecting average student-teacher and student-staff ratio data. With large schools, it is unlikely for each student, teacher, or other staff member to be in close contact with all other individuals. Instead, for each individual in the school layer we model their close and effective contacts as a subset of contacts from their school who can infect them by sampling a random set of *n* other individuals in their school, where *n* is drawn from a Poisson distribution with rate parameter $\lambda_s$ equal to the average class size ($\lambda_s$ = 20 as a default).

**2.4.2.3 *Workplaces and community*.** The labor force is drawn using employment rates by age, and non-teachers are assigned to workplaces using data on establishment sizes. Workers are assigned to workplaces using a similar method with an initial reference worker sampled

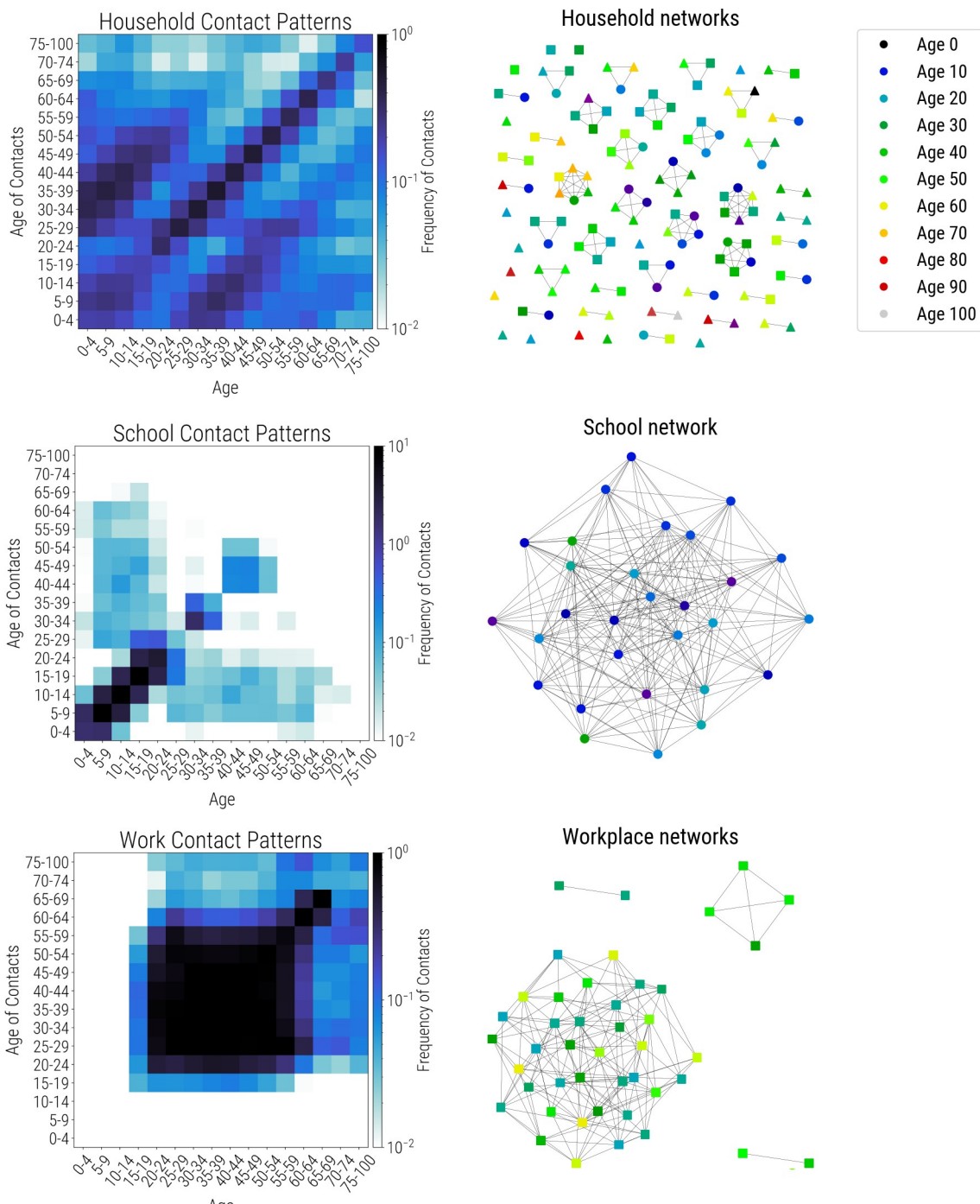

**Fig 4.** Synthetic population networks for households (top), schools (middle), and workplaces (bottom). Age-specific contact matrices are shown on the left, while actual connectivity patterns for a 127-person subsample of a population of 10,000 individuals are shown on the right. All individuals are present in the household network, including some with no household connections. A subset of these individuals, including teachers, are present in the school network (circles); another subset is present in workplace networks (squares); some individuals are in neither school nor work networks (triangles).

from the labor force and their co-workers inferred from age mixing patterns within the work-force. All workers (teachers included) are drawn at random from the population, to reflect the general mixing of adults from different neighborhoods at work. Similar to the school layer,

large workplaces are unlikely to be fully connected graphs of contacts. Instead, for each worker, we model their close contacts as a subset of $n$ contacts from other individuals in their workplace, where $n$ is drawn from a Poisson distribution with rate parameter $\lambda_w$ equal to the estimated maximum number of close contacts in the workplace ($\lambda_w = 20$ as a default).

For contacts in the general community, we draw $n$ random contacts for each individual from other individuals in the population, where $n$ is drawn from a Poisson distribution with rate parameter $\lambda_c$ equal to the expected number of contacts in the general community (with $\lambda_c = 20$ as a default, as above). Connections in this layer reflect the nature of contacts in shared public spaces like parks and recreational spaces, shopping centers, community centers, and public transportation. All links between individuals are considered undirected to reflect the ability of either individual in the pair to infect the other.

The generated multilayer network of household, school, work, and community network layers represents a population with realistic microstructure. This framework can also be extended to consider more detailed interactions in key additional settings, such as hospitals, encampments, shelters for those experiencing homelessness, and long term care facilities.

**2.4.3 Hybrid networks.** Covasim contains a third option for generating contact networks, which captures some of the realism of the SynthPops approach but does not require as much input data, and is more readily adaptable to other settings. As such, it is a "hybrid" approach between a fully random network and a fully data-derived network. As with SynthPops, each person in the population has contacts in their household, school (for children), workplace (for adults), and community. A population of individuals is generated according to a location-specific age distribution, and each individual is randomly assigned to a household using location-specific data on household sizes (using the pre-loaded UN data described above).

Unlike SynthPops, the hybrid algorithm does not account for the distribution of ages within a household. Children are assigned to schools and adults to workplaces, each with a user-specified number of fixed daily contacts (by default, Poisson-distributed with means of 20 for schools and 16 for workplaces, chosen to match the mean values for SynthPops networks). Individuals additionally have contacts with others in the community (by default, Poisson distributed with a mean of 20). All children and young adults aged between 6 and 22 are assigned to schools and universities, and all adults between 22 and 65 are assigned to workplaces. This distinguishes it from SynthPops where enrollment or employment varies depending on the given data. A comparison of the different population structure options available in Covasim is listed in Table 3.

**Table 3. Comparison of population options in Covasim.**

| Population type | Data requirements | Best suited for | Not well suited for |
|---|---|---|---|
| Random networks | None | Models of transmission in special settings such as prisons or cruise ships | Large or complex populations |
| Hybrid networks | Data on the age distribution and household sizes for each country are pre-loaded<br>No additional data are required, but users can optionally specify the daily number of school, workplace, and community contacts | Population network models in data-rich settings; adaptable and suited to most modeling contexts | Populations with high heterogeneity in contact patterns or size distributions |
| SynthPops networks | Household, school, workplace, and community age mixing patterns<br>School size distributions, enrollment rates by age, student-teacher ratios<br>Workplace size distributions, employment rates by age<br>Number of households, size distribution, and age/sex distribution | Complex populations in data-rich settings | Settings where the data requirements cannot be met, or where other social settings are critical contexts for disease transmission |

## 2.5 Interventions

A core function of Covasim is modeling the effect of interventions on disease transmission or health outcomes, to understand the impact that different policy options may have in a specific setting. In general, interventions are modeled as changes to parameter values. Covasim has built-in implementations of the common interventions described below, as well as the ability for users to create their own interventions, which can either be derived from the base intervention class, or be simple functions that modify the simulation object. Both built-in and user-defined interventions have full access to the simulation object at each timestep, which means that user-defined interventions can dynamically modify any aspect of the simulation. This can be used to create interventions more specific than those included by default in Covasim, such as age-specific physical distancing or quarantine, or interventions that are dynamically "triggered" based on the current or past state of the simulation.

**2.5.1 Physical distancing, masks, and hygiene.**   The most basic intervention in Covasim is to reduce transmissibility ($\beta$) starting on a given day. This can be used to reflect both (a) reductions in transmissibility per contact, such as through mask wearing, personal protective equipment, hand-washing, and maintaining physical distance; and (b) reductions in the number of contacts at home, school, work, or in the community. However, Covasim also includes an "edge-clipping" intervention (considering a contact between two agents as a weighted "edge" between two "nodes"), where $\beta$ remains unchanged but the number of contacts that person has is reduced. Complete school and workplace closures, for example, can be modeled either by setting $\beta$ to 0, or by removing all edges in those contact layers; partial closures can be modeled by smaller reductions in either $\beta$ or the number of contacts.

In general, both types of interventions have similar impact–for example, halving the number of contacts and keeping $\beta$ constant will produce very similar epidemic trajectories as halving $\beta$ and keeping the number of contacts constant. However, the distinction becomes important when considering the interaction between physical distancing and other interventions. For example, in a contact tracing scenario, the number of contacts who require tracing, number of tests performed, and number of people placed in quarantine are all strongly affected by whether physical distancing is implemented as a reduction in $\beta$ of specific edges, or removing those edges entirely.

**2.5.2 Testing and diagnosis.**   Testing can be modeled in two different ways within Covasim, depending on the format of testing data and purpose of the analysis. The first method allows the user to specify the probabilities that people with different risk factors and levels of symptoms will receive a test on each day. Separate daily testing probabilities can be entered for those with/without symptoms, and those in/out of quarantine. The model will then estimate the number of tests performed on each day. The second method allows the user to enter the number of tests performed on each day directly, including multipliers on the probability of a person receiving a test if they have symptoms, are in quarantine, or are over a certain age. This method will then allocate the tests among the population. If data on the number of daily tests performed each day are available, the second method is preferable.

Once a person is tested, the model contains a delay parameter that indicates how long people need to wait for their results, as well as a loss-to-follow-up parameter that indicates the probability that people will not receive their results. Additional parameters control the sensitivity and specificity of the tests.

**2.5.3 Contact tracing.**   Contact tracing corresponds to notifying individuals that they have had contact with a confirmed case, so that they can be quarantined, tested, or otherwise change their behavior. Contact tracing in Covasim is parameterized by the probability that a contact can be traced, and by the time taken to identify and notify contacts. Both parameters can vary

by type of contact, and can be controlled by the user. For example, it may be reasonable to assume that people can trace members of their household immediately and with 100% probability, while tracing work colleagues may take several days and may be incomplete. Digital contact tracing can be approximated in Covasim as a standard contact-tracing intervention with zero delays, with the caveat that tracing multiple steps (i.e., contacts of contacts) within a single day would require a custom intervention.

**2.5.4 Isolation of positives and contact quarantine.** Isolation (referring to behavior changes after a person is diagnosed with COVID-19) and quarantine (referring to behavior changes after a person is identified as a known contact of someone with confirmed or suspected COVID-19) are the primary means by which testing interventions reduce transmission. In Covasim, people diagnosed with COVID-19 can be isolated. Their contacts who have been traced can be placed in quarantine with a specified level of compliance; people in quarantine may also have an increased probability of being tested. People in isolation or quarantine typically have a lower probability of infecting others (if infectious) or of acquiring COVID-19 (if quarantined and susceptible). The default reductions for isolation are 70% in the household and 90% in school, work, and community layers, while quarantine is assumed to have lower compliance (40% reduction in the household and 80% in other layers). However, if psychosocial support is not provided to people in home isolation or quarantine, there may be an increased risk of passing on infection to, or acquiring infection from, other household members. For performance reasons, isolation and quarantine are implemented as reductions in per-contact transmission risk rather than changes in the number of contacts; for realistic parameter values (i.e., $\beta \ll 1$), the difference between these implementations should be negligible.

**2.5.5 Vaccines and treatments.** Pharmaceutical interventions, especially vaccines, are an increasingly important part of public health responses to COVID-19. However, there are significant modeling challenges due to the large number of vaccine candidates under investigation, coupled with the considerable uncertainty regarding their properties–such as the extent to which they block acquisition and transmission as well as symptoms, how much protection is conferred by a single dose, the extent to which immunity wanes over time, and their effectiveness against different COVID-19 strains [70]. Vaccines in Covasim are modeled by adjusting individuals' susceptibility to infection and probability of developing symptoms after being infected; both of these modifications affect the overall probability of progressing to severe disease and death. Additional flexibility, including waning efficacy and differential effectiveness across variants, will be incorporated as trial results become available. Though treatments for COVID-19 have so far had only modest results in clinical trials [71], they can be implemented in Covasim as interventions that reduce the probability of progressing to severe disease or death.

## 2.6 Additional features

**2.6.1 Data inputs.** In addition to the demographic and contact network data available via SynthPops, Covasim includes interfaces to automatically load COVID-19 epidemiology data, such as time series data on deaths and diagnosed cases, from several publicly available databases. These databases include the Corona Data Scraper (coronadatascraper.com), the European Centre for Disease Prevention and Control (ecdc.europa.eu), and the COVID Tracking Project (covidtracking.com). At the time of writing, these data are available for over 4,000 unique locations, including most countries in the world (administrative level 0), all US states and many administrative level 1 (i.e., subnational) regions in Europe, and some administrative level 2 regions in Europe and the US (i.e., US counties).

**2.6.2 Dynamic rescaling.**   One of the major challenges with agent-based models is simulating a sufficient number of agents to capture an epidemic at early, middle, and late stages, without requiring cumbersome levels of memory or processor usage. Whereas compartmental SEIR models require the same amount of computation time regardless of the population size being modeled, the performance of agent-based models typically scales linearly or supralinearly with population size (see Section 2.7.1). As a consequence, many agent-based models, including Covasim, include an optional "scaling factor", where a single agent in the model is assumed to represent multiple people in the real world. A scaling factor of 10, for example, corresponds to the assumption that the epidemic dynamics in a city of 2 million people can be considered as the sum of the epidemic dynamics of 10 identical subregions of 200,000 people each.

However, the limitation of this approach is that it introduces a discretization of results: model outputs can only be produced in increments of the scaling factor, so relatively rare events, such as deaths, may not have sufficient granularity to reflect the epidemic behavior at a small scale. In addition, using too few agents in the model introduces stochastic variability patterns that do not reflect real-world processes in the entire population.

To circumvent this, Covasim includes an option for dynamic rescaling. Initially, when the epidemic is small, there is no scaling performed: one agent corresponds to one person. Once a certain threshold is reached, however (by default, 5% of the population is non-susceptible), the non-susceptible agents in the model are downsampled and a corresponding scaling factor is introduced (by default, a factor of 1.2 is used). For example, in a simulation of 100,000 agents representing a true population of 1 million with a threshold of 10% and a rescaling factor of 2, dynamic rescaling would be triggered when cumulative infections surpass 10,000, leaving 90,000 susceptible agents; dynamic rescaling would then resample the non-susceptible population to 5,000 (now representing 10,000 people) and increase the number of susceptible agents to 95,000 (now representing 190,000 people), i.e. with every agent now counting as two. If the epidemic expands further, this process will repeat iteratively until the scale factor reaches its upper limit (which in this example is 10, and which would be reached after 100,000 cumulative infections). Through this process, arbitrarily large populations can be modeled, even starting from a single infection, maintaining a constant level of precision and computation time throughout.

**2.6.3 Health system capacity.**   Individuals in the model who have severe and critical symptoms are assumed to require regular and intensive care unit (ICU) hospital beds, respectively, including ventilation in the latter case. The number of available hospital beds (ICU and otherwise) beds are input parameters. If the model estimates that the number of severe/critical cases is greater than the number of available non-ICU/ICU beds, then the health system capacity is exceeded. This means that severely ill individuals have an increased probability of progressing to critical, and critically ill individuals who are unable to access treatment have an increased mortality rate (by default, both by a factor of 2).

**2.6.4 Importations.**   The spatial movement of agents is not currently modeled explicitly in Covasim, and the population size for a given simulation is fixed. Thus, importations are implemented as spontaneous infections among the susceptible population. This corresponds to agents who become infected elsewhere and then return to the population.

**2.6.5 Model outputs.**   By default, Covasim outputs three main types of result: "stocks" (e.g., the number of people with currently active infections on a given day), "flows" (e.g., the number of new infections on a given day), and "cumulative flows" (e.g., the cumulative number of infections up to a given day). For states that cannot be transitioned out of (e.g. death, plus recovery if reinfection is not considered), the stock is equal to the cumulative flow. Flows that are calculated in the model include: the number of new infections and the number of people

who become infectious on that timestep; the number of tests performed, new positive diagnoses, and number of people placed in quarantine; the number of people who develop mild, severe, and critical symptoms; and the number of people who recover or die. The date of each transition (e.g., from critically ill to dead) is also recorded. By default, these results are summed over the entire population on each day; results for subpopulations can be obtained by defining custom analyzers, as described in Section 2.6.7.

**2.6.6 Reproduction number and doubling time.** In addition to these core outputs, Covasim includes several outputs for additional analysis. For example, several methods are implemented to compute the effective reproduction number $R_e$. Numerous definitions of $R_e$ exist; in standard SIR modeling, the most common definition ("method 1") is [72]:

$$R_e = R_0 S/N$$

where $R_0$ is the basic reproduction number, $S$ is the number of susceptibles, and $N$ is the total population size. However, with respect to COVID-19, many authors instead define $R_e$ to include the effects of interventions, due to the implications that $R_e = 1$ has for epidemic control.

A second common definition of $R_e$ ("method 2") is to first determine the total number of people who became infectious on day $t$, then count the total number of people these people went on to infect, and then divide the latter by the former. "Method 3" is the same as method 2, except it counts the number of people who stopped being infectious on day $t$ (i.e., recovered or died), and then counts the number of those people infected. Unlike in a compartmental model, where $R_e$ can only be estimated by using simplifying assumptions, in an agent-based model, methods 2 and 3 can be implemented by simply counting exactly how many secondary infections are caused by each primary infection. By doing so, all details of the epidemic–including time-varying viral loads, population-level and localized immunity, interventions, network factors, and other effects–are automatically incorporated, and do not need to be considered separately.

While methods 2 and 3 are implemented in Covasim, they have the disadvantage that they introduce significant temporal blurring, due to the potentially long infectious period (and, for method 3, the long recovery period). To avoid this limitation, the default method Covasim uses for computing $R_e$ is to divide the number of new infections on day $t$ by the number of actively infectious people on day $t$, multiplied by the average duration of infectiousness ("method 4"). This definition of $R_e$ is nearly identical to the definition of the "instantaneous reproductive number" in Gostic et al. [73], which in that study is used as the ground truth against which other $R_e$ estimators are compared.

Covasim also includes an estimate of the epidemic doubling time, computed similarly to the "rule of 69.3" [74], specifically:

$$T = \frac{w\log(2)}{\log(n_i(t)/n_i(t - w))}$$

where $T$ is the doubling time, $w$ is the window length over which to compute the doubling time (3 days by default), and $n_i(t)$ is the cumulative number of infections at time $t$.

**2.6.7 Analyzers.** In addition to interventions, Covasim also includes a library of "analyzers". Like interventions, in principle they can access and modify any aspect of the simulation state. However, they are typically used to record additional details about the internal state of the model that are not included as standard outputs (e.g., the age distribution of infections at a given point in time). By convention, interventions and analyzers differ in that interventions modify the state of the simulation (and are applied at the beginning of each timestep), while analyzers record the state (and are applied at the end of each timestep).

**2.6.8 Calibration.** The process of calibration involves finding parameter values that minimize a function that measures the difference between observed data (which typically includes daily confirmed cases, hospitalizations, deaths, and number of tests conducted) and the model predictions. Since most data being calibrated to are time series count data, this function is defined as:

$$L = \sum_s \sum_t w_s f(c_d^{s,t}, c_m^{s,t})$$

where $s$ refers to the type of data observed (such cumulative confirmed cases or number of deaths); $t$ is the time index; $w_s$ is the weight associated with $s$; $c_d^{s,t}$ and $c_m^{s,t}$ are the counts from the data and model, respectively, for this time series at this time index; and $f$ is the loss, objective, or goodness-of-fit function (e.g., normalized absolute error, mean absolute error, mean squared error, or the Poisson test statistic [75]). By default, Covasim calculates the loss using normalized absolute error. Depending on underlying distributional assumptions, minimizing the normalized absolute error can sometimes give parameter estimates that are equivalent to the estimates that maximize the log-likelihood (or an approximation thereof, as in approximate Bayesian computation [76]). Intuitively, most distributional assumptions mean that larger errors imply a lower log-likelihood. However, we do not make explicit distributional assumptions, so caution is advised with treating them as statistically rigorous likelihoods.

Calibrating any model to the COVID-19 epidemic is an inherently difficult task: not only is there significant uncertainty around the reported data, but there are also many possible combinations of parameter values that could give rise to these data. Thus, in a typical calibration workflow, most parameters are fixed at the best available values from the literature, and only essential parameters (for example, $\beta$) are allowed to vary.

Calibration is often performed externally to Covasim. However, since a single model run returns a scalar loss value, these runs can be easily integrated into standardized calibration frameworks. Any standard optimization library–such as the optimization module of SciPy–can be easily adapted (as long as it can handle stochastic results, which standard gradient descent cannot), as can more advanced methods such as the adaptive stochastic descend method of the Sciris library [77], or Bayesian approaches such as history matching [78] and sequential Monte Carlo methods [79]. To date, the Optuna hyperparameter optimization library [80] has proven to be the most effective approach for calibration, and an implementation is included in the codebase.

## 2.7 Software architecture

Covasim was developed for Python 3.8 using the SciPy (scipy.org) ecosystem [81]. It uses NumPy (numpy.org), Pandas (pandas.pydata.org), and Numba (numba.pydata.org) for fast numerical computing; Matplotlib (matplotlib.org) and Plotly (plotly.com) for plotting; and Sciris (sciris.org) for data structures, parallelization, and other utilities.

The source code for Covasim is available via both the Python Package Index (via pip install covasim) and GitHub (github.com/institutefordiseasemodeling/covasim). Covasim is fully open-source, released under the Creative Commons Attribution-ShareAlike 4.0 International Public License. More information is available at covasim.org, with full documentation and a comprehensive set of tutorials available at docs.covasim.org.

**2.7.1 Performance.** All core numerical algorithms in the Covasim integration loop–specifically, calculating intra-host viral load, per-person susceptibility and transmissibility, and which contacts of an infected person become infected themselves–are implemented as highly optimized 32-bit array operations in Numba. For further efficiency, agents are not represented

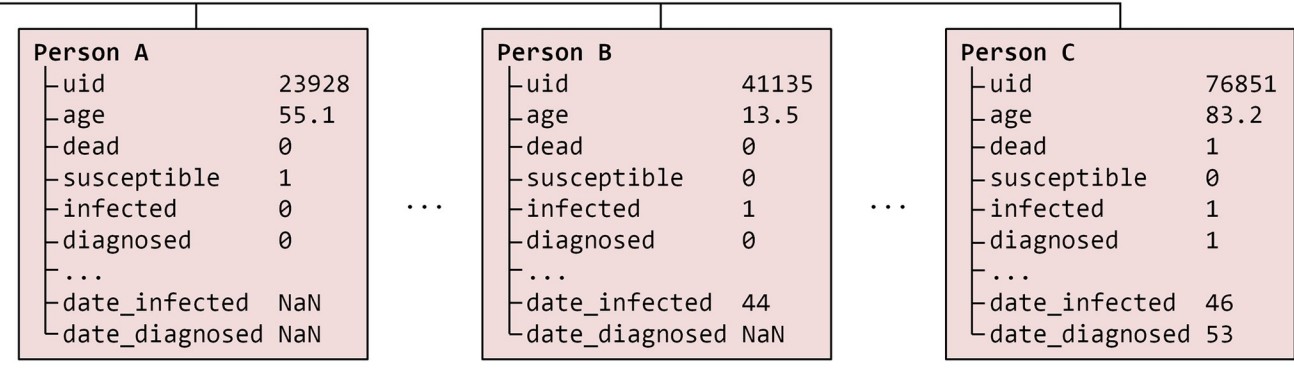

**Fig 5.** Illustration of the standard object-oriented approach for implementing agent-based models (top), where each agent is a separate object, compared with the approach used in Covasim (bottom), where agents are represented as slices through a set of state arrays. Dots (. . .) represent omitted entries. In practice, each agent has several dozen states, and there are typically hundreds of thousands of agents.

as individual objects, but rather as indices of one-dimensional state arrays (Fig 5). This avoids the need to use an explicit for-loop over each agent on every integration timestep, increasing performance by more than an order of magnitude. Similarly, contacts between all agents in the model are stored as a single array of "edges" per contact layer.

As shown in Fig 6, these software optimizations allow Covasim to achieve high levels of performance, despite being implemented purely in Python. Performance scales linearly with population size over multiple orders of magnitude: memory scales at a rate of roughly one agent per 1 KB of memory, while single-core compute time (benchmarked on an Intel i9-8950HK laptop processor) scales at a rate of roughly 7 million simulated person-days per second of CPU time. These speed and memory use results are comparable to OpenABM-Covid19, despite the latter being implemented in C [23]. One consequence of the array-based implementation is that compute time depends on the number of agents and the number of connections per agent, but is independent of the number of infected agents; this is because uninfected agents are simply represented as zeros in the transmission probabilities vector.

Due to Covasim's computational efficiency, it is feasible to run realistic scenarios, such as tens of thousands of infections among a susceptible population of hundreds of thousands of people for a duration of 12 months, in under a minute on a personal laptop. Covasim is also suited to high-performance computing environments, with support for parallelization via the built-in "multiprocessing" library; it can also be adapted easily to other parallel processing

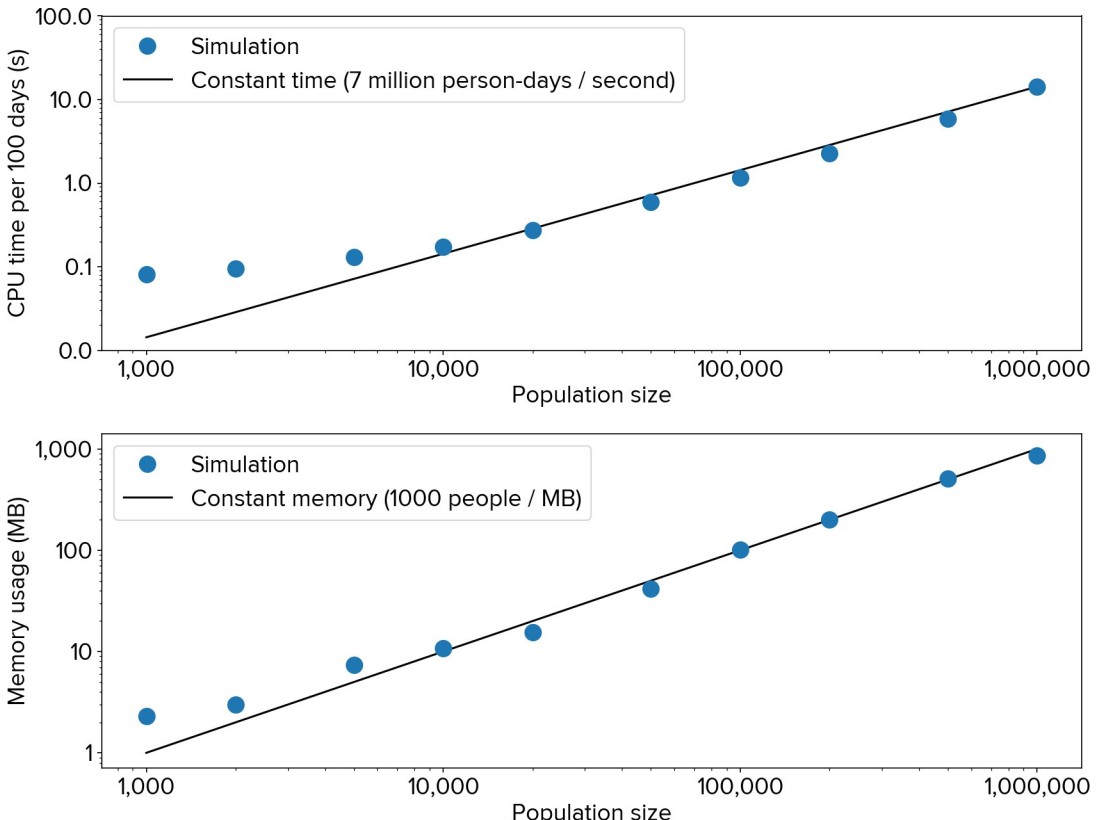

**Fig 6.** Covasim performance in terms of processor usage (top) and memory usage (bottom), for the number of agents shown, simulated for 100 days. There is roughly linear scaling over three orders of magnitude of population size.

libraries such as Celery and Dask. Although in some special situations it is possible to split a single simulation across multiple cores, parallel processing is used primarily to run multiple independent simulations simultaneously, such as for uncertainty analyses or calibration.

**2.7.2 Deployment and access.** While Covasim is primarily intended to be used via Python scripts, a number of other options for using it are also available. A simple webapp for Covasim has been developed, based on Vue.js (for the frontend), ScirisWeb (for communicating between the frontend and the backend), Flask (for running the backend), and Gunicorn/NGINX (for running the server); this webapp is available at app.covasim.org. A screenshot of the user interface is shown in Fig 7. A pre-built version of Covasim, including the webapp, is also available on Docker Hub (hub.docker.com). Covasim can also be run via R using the "reticulate" library, and from the command line via the "fire" library.

**2.7.3 Software tests.** Covasim includes an extensive suite of both integration tests and unit tests; code coverage for version 2.1.1 is 89%. In addition, outputs from the default simulations for each version are compared against cached values in the repository; since random seeds are stored, results are exactly reproducible despite the stochasticity in the model. When new data become available and parameter values are updated, previous parameters are stored, ensuring that any changes affecting the model outputs are intentional, and that previous versions can be easily retrieved and compared against. The test suite includes unit tests (e.g., checking that sampling functions produce the specified distributions; that simulations loaded from file exactly match the original), functional tests (e.g., that a simulation run with a

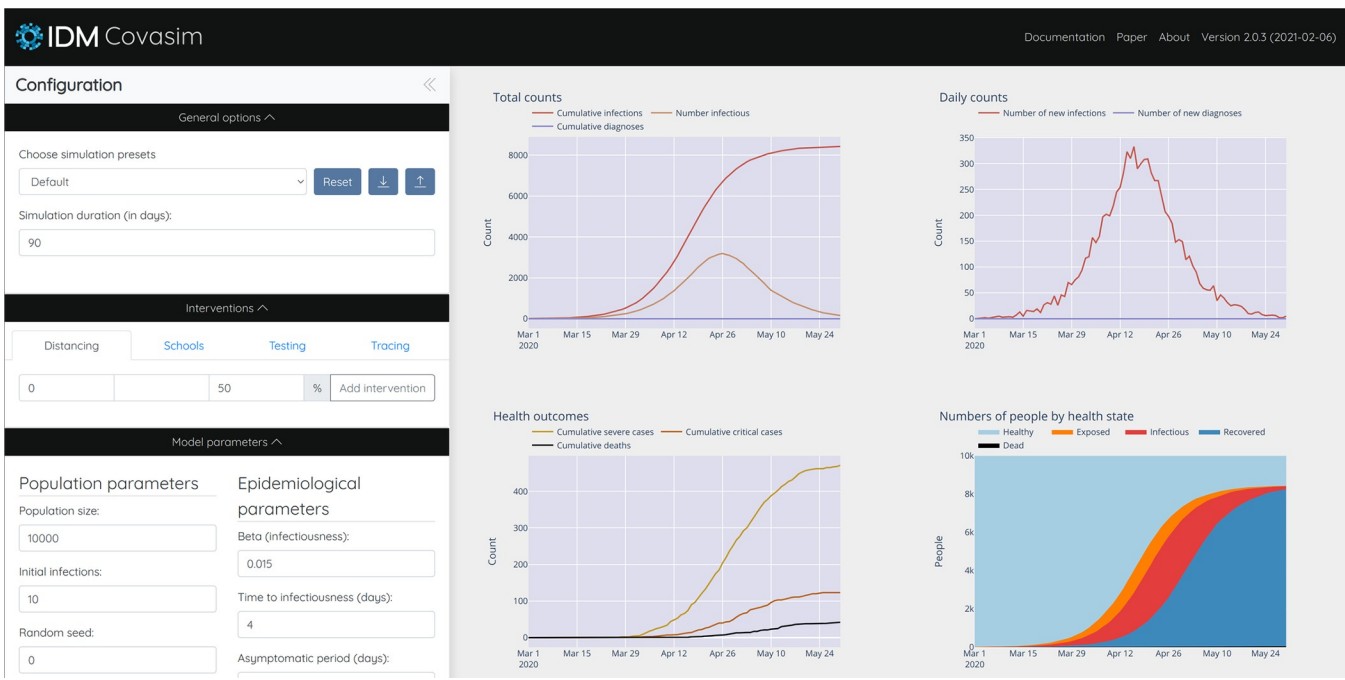

**Fig 7. Covasim webapp user interface; screenshot taken from http://app.covasim.org.**

particular analyzer produces a plot), and end-to-end "scientific" tests (e.g., that an increase in mortality rate leads to more deaths, while adding NPIs leads to fewer).

# 3 Results

## 3.1 Example usage

Several of Covasim's standard features are illustrated in Fig 8A. It represents a simulation of 200,000 people, from February 10 to June 29 2020, starting with 75 seed infections. After an initial 45 days of uncontrolled epidemic spread, the following interventions are applied: March 26, close schools and reduce work and community contacts to 70% of their original values; April 10, reduce work and community to 30% of their original values; May 5, reopen work and community to 80% of their original values; May 20, begin testing 10% of people with COVID-like illness each day, and trace the contacts of people who test positive.

By default, Covasim shows time series for key cumulative counts, daily counts, and health outcomes (including deaths). All plotting outputs are configurable, and results can also be saved in Excel, JSON, or NumPy formats for further processing. While a full Covasim application would likely include additional complexity regarding calibration and plotting, other aspects of the example shown in Fig 8A are comparable to a real-world exploratory policy analysis. Despite this, the Python script used to generate Fig 8A is only 28 lines; this code is listed in Fig 8B.

In addition to running single simulations, Covasim also allows the user to run multiple simulations, which can be averaged over to determine forecast intervals. By default, the "80% forecast interval" is used, i.e. between the 10th and 90th percentiles of the simulated trajectories. Since these forecast intervals are typically produced by a combination of both stochastic variability ("aleatory uncertainty") and imperfect knowledge of the "true" parameter values

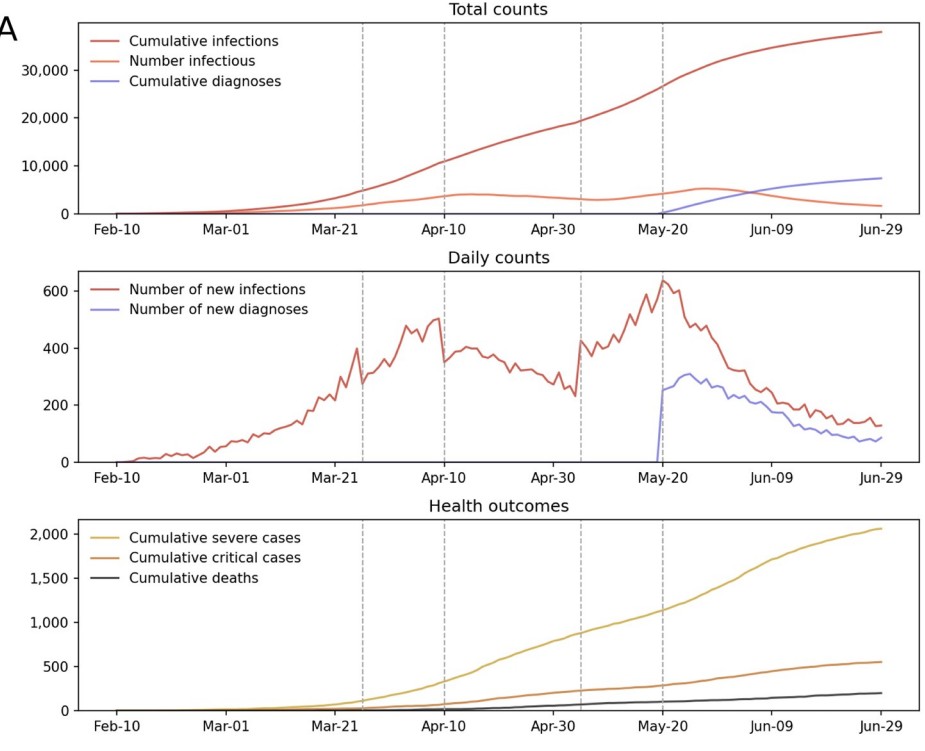

**Fig 8.** (A) Illustrative example of a single run of a Covasim simulation. Interventions (described in the text) are shown as dashed vertical lines. (B) Full listing of the code for this simulation, including defining the parameters of the simulation (lines 4–11); defining the interventions (lines 14–23); and creating, running, and plotting the simulation (lines 26–28).

("epistemic uncertainty"), they should not be interpreted as statistically rigorous Bayesian credible intervals [82,83]. Multiple simulations can also be used to quickly run different scenarios in parallel and compare them, as shown in Fig 9.

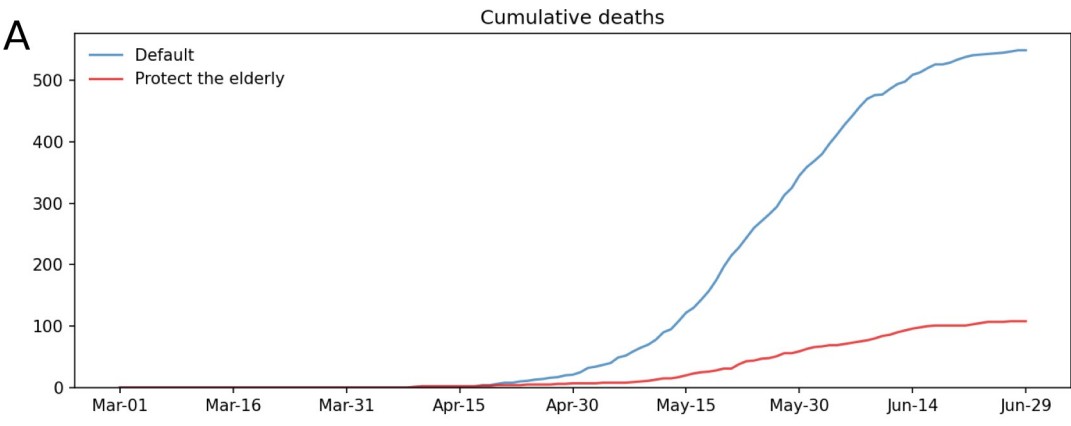

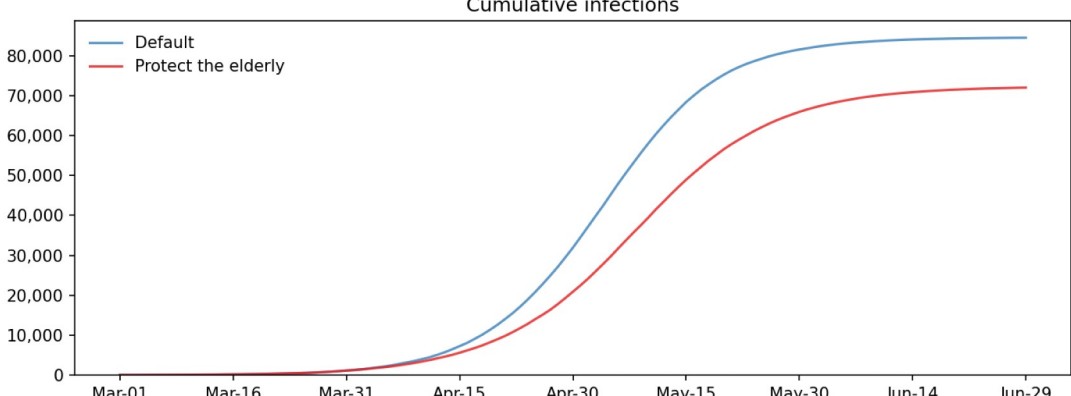

```
B    1  import covasim as cv
     2
     3  def protect_elderly(sim):
     4      if sim.t == sim.day('2021-04-01'):
     5          elderly = sim.people.age>70
     6          sim.people.rel_sus[elderly] = 0.0
     7
     8  pars = {'pop_size':100e3, 'start_day':'2021-03-01', 'n_days':120}
     9  s1 = cv.Sim(pars, label='Default')
    10  s2 = cv.Sim(pars, label='Protect the elderly', interventions=protect_elderly)
    11  cv.MultiSim([s1, s2]).run().plot(to_plot=['cum_deaths', 'cum_infections'])
```

**Fig 9.** (A) Illustrative example of a scenario comparison using a simple custom intervention ("protecting the elderly", i.e. removing all transmission among people over age 70 after a certain date). (B) Full listing of the code for this simulation, showing the intervention definition (lines 3–6), and a compact way of creating the simulations, running them in parallel, and plotting them (lines 8–11).

## 3.2 Transmission analyses

The preceding examples illustrate some aspects of Covasim's core functionality that are used in most applications. More in-depth analyses are also possible, leveraging either the default outputs, or the fact that the full state of the model is accessible to the user at every timestep via custom analysis functions.

For example, detailed information about the transmission tree is stored for each simulation. This information can be used to determine the detailed microstructure of the infection patterns in a given simulation. Complete transmission trees for a small network under three

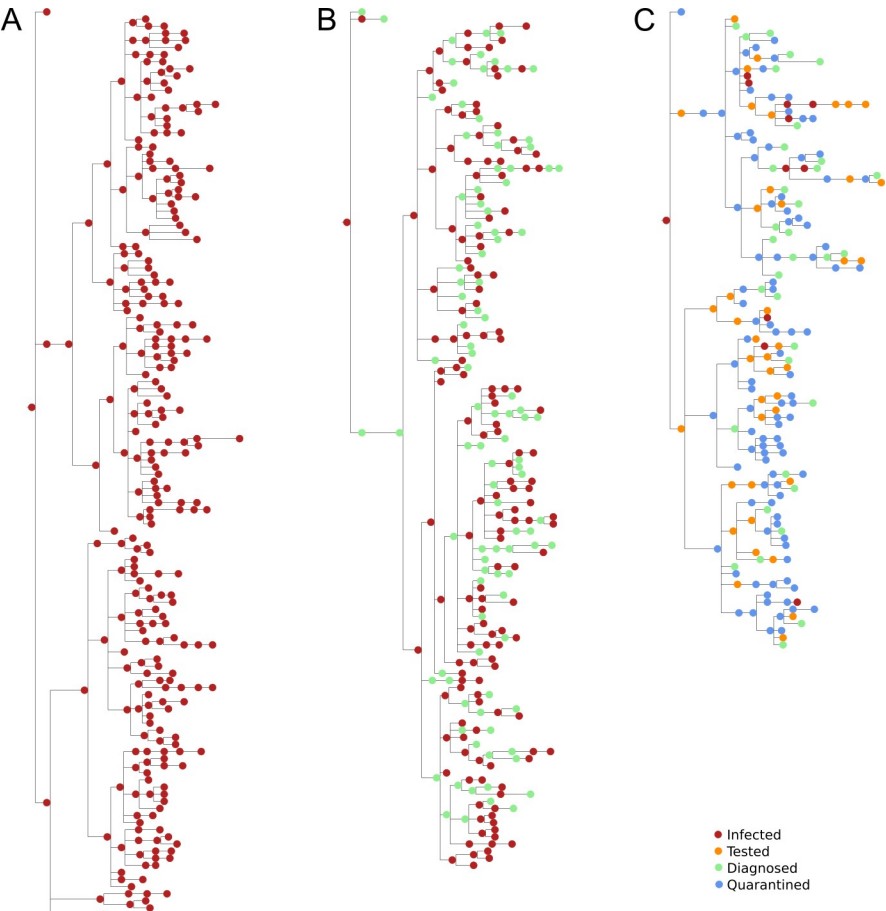

**Fig 10.** Example transmission trees for a hypothetical population of 300 individuals with a single seed infection on day 1, with (A) no interventions, (B) testing only, and (C) testing plus contact tracing. Time is shown on the horizontal axis, with each tree representing approximately 90 days. The vertical size of each tree is proportional to the total number of infections.

different intervention scenarios are shown in Fig 10, visualized via the ETE Toolkit [84]. For realistically sized networks, it is not feasible to visualize entire transmission trees. However, their statistical properties can be analyzed to determine transmission routes and potential intervention targets. For example, such information can be used to determine the net contribution of schools (or even teachers at schools) to the overall epidemic trajectory [27].

## 3.3 Case study

Here we provide a case study of how Covasim was used to inform a policy decision in King County (the local government area that includes the city of Seattle), Washington, USA; a full description of the methodology used is given in [25]. Briefly, we used Optuna to calibrate Covasim to epidemiological and program data from January 27 to November 14 2020; these data are available from the Public Health Seattle King County data dashboard [3]. We then ran the model with eight different calibrated parameter sets (with multiple parameter sets used to capture parametric uncertainty) to (a) estimate unobserved quantities, such as the number of new infections and the case detection rate; (b) estimate the impact of proposed new mobility restrictions (such as limiting indoor dining) scheduled to start on November 16, which we

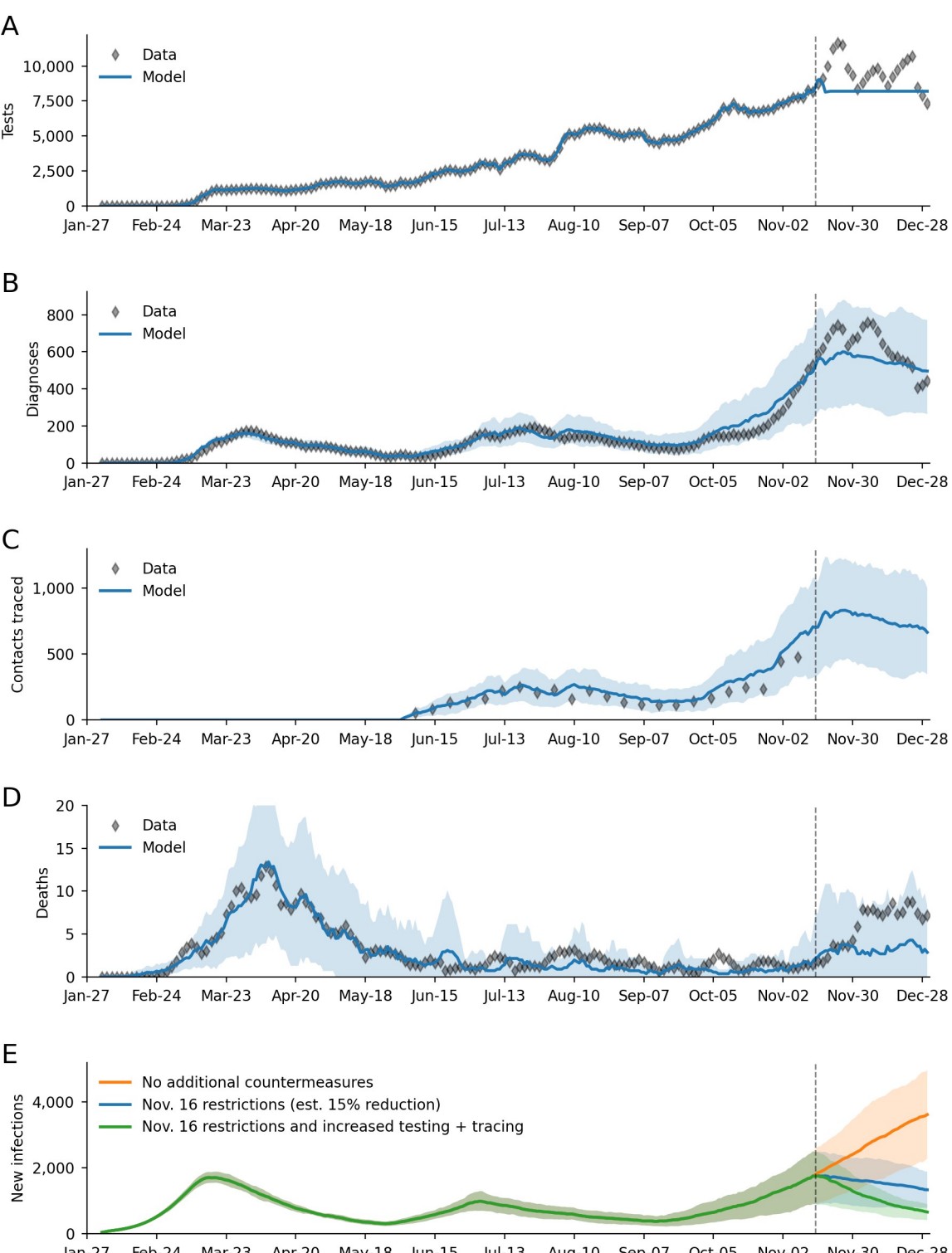

**Fig 11. Example calibration of Covasim to data from Seattle/King County, Washington, USA from January 27 to November 14 2020 (dashed line), with projections until December 31, including additional restrictions imposed on November 16.** (A) Number of daily COVID-19 tests, which are used as input data. (B) Calibration to the number of daily COVID-19 diagnoses. (C) Calibration to the number of daily contacts traced (weekly averages shown; data past prediction date are not available). (D) Calibration to the number of daily COVID-19 deaths. (E) Projections of the number of new infections if restrictions had not been implemented, with the restrictions as implemented, and if restrictions were implemented together with increases in testing and contact tracing. Bands show 80% forecast intervals; data are rolling 7-day averages to account for weekend reporting delays.

estimated would result in a 15% reduction in transmission [85]; and (c) compare this scenario with counterfactual scenarios of either not implementing the scheduled restrictions, or by implementing them together with increased testing and contact tracing.

As shown in Fig 11, Covasim was able to capture numerous features of the epidemic during the calibration period, including numbers of tests and contacts traced (which were used as input data, along with mobility data from SafeGraph; see safegraph.com); the three infection "waves" (spring, summer, and fall); changes in test positivity rate (not shown), and numbers of deaths. During the scenario period, we assumed that the number of tests conducted per day would remain constant at the average value from the previous 7 days (Fig 11A).

Despite a rapid increase of cases in the preceding weeks, the model predicted counterintuitively that even these modest mobility restrictions would be sufficient to stop the rise in cases (Fig 11B), a projection that turned out to be accurate. (Note that using actual testing data for this period, rather than assuming a constant number of tests, would have resulted in an even more accurate prediction of diagnoses, though of course these data were not available at the time the prediction was made). While the model correctly predicted the trend in cases, it underestimated the number of deaths (Fig 11D), although the observations were still within the 80% forecast interval (the large uncertainty interval for deaths is a consequence of the small numbers of events being predicted, i.e., fewer than 10 deaths per day; this forecast interval includes both parametric and stochastic uncertainty, as described in Section 3.1). This underestimate was likely due to assuming a continuation of infection patterns that occurred over the summer and early fall, during which younger adults were disproportionately infected compared to older ones.

Finally, we predicted that had the additional restrictions not been implemented, by the end of the year, daily infection rates would have been roughly three times as high as actually occurred (Fig 11E). Had testing and contact tracing programs been rapidly scaled up (by 50% and five fold respectively), we estimated the number of infections would have been approximately halved. These predictions helped provide quantitative support for public health decisions regarding mobility restrictions and increased testing.

## 4 Discussion

The COVID-19 pandemic has presented an unprecedented challenge to the disease modeling community in terms of requiring rapid, accurate predictions, often based on extremely limited data, with consequences of global scale. Covasim was developed to help policymakers make decisions based on the best available data, while taking into account the large uncertainties that remain in terms of COVID-19 transmission dynamics, disease progression, and other aspects of its biology, such as the proportions of asymptomatic and presymptomatic transmission.

We prioritized five different factors when developing Covasim: rapid development process, computational performance, flexibility, simplicity for users, and simplicity for developers. Striking a balance between these factors required making certain tradeoffs. For example, choosing to implement Covasim in Python instead of C++ or Java significantly reduced development time and increased simplicity for users and developers; however, it imposed a large penalty on performance. While we were able to solve this by using Numba and vectorized state arrays in place of object-oriented agents, this implementation increased development time and increased the complexity for developers. Another tradeoff we encountered is that while the gold standard in simplicity of use remains interactive webapps [86], the limited flexibility such webapps provide means that most Covasim users to date have instead used Python scripts to run analyses.

Beyond implementation tradeoffs, it is worth noting that in many cases, compartmental models offer simpler, faster, and more robust results than agent-based models such as Covasim. Indeed, many of the most influential COVID-19 models that have been developed to date have been compartmental models [4,85,87,88]. However, compartmental models have two major limitations. First, they cannot be easily adapted to changing epidemic conditions, such as new strains or multiple types of vaccine, since these often require a combinatorial explosion in the number of compartments [89,90]. Second, they are unsuitable for answering questions that depend on details of behavior at the individual level, such as superspreading events, transmission within multigenerational households, school classroom cohorting, and contact tracing. While it is possible to approximate some of these phenomena in compartmental models [91,92], these approximations typically exclude important factors such as time delays. Some of the issues regarding compartmental models' predictive performance [93–95] may be partly a consequence of their inability to capture key mechanisms of epidemic spread. While agent-based models, including Covasim, are difficult to deploy widely enough, and calibrate quickly enough, to be a feasible replacement for compartmental models, they can provide a mechanistic understanding of the COVID-19 epidemic in ways that compartmental models cannot.

### 4.1 Limitations of Covasim

Covasim is subject to the usual limitations of mathematical models, most notably constraints around the degree of realism that can be captured. For example, human contact patterns are intractably complex, and the algorithms that Covasim uses to approximate these are necessarily quite simplified.

Like all models, the quality of the outputs depends on the quality of the inputs, and many of the parameters on which Covasim relies are still subject to large uncertainties. Most critically, the proportion of asymptomatics and their relative transmission intensity, and the proportion of presymptomatic transmission, strongly affect the number of tests required in order to achieve workable COVID-19 suppression via testing-based interventions.

Dynamical models are commonly validated by comparing their projections against data on what actually happened, as shown in the case study (Fig 11). However, several challenges are commonly encountered when using this approach for COVID-19, including (a) data quality issues (such as low case detection rates and under-reporting of deaths); (b) the difficulty of predicting future social and political responses that would significantly impact model projections (such as the timing of school and workplace reopening, or a sudden increase in testing rates, as in the case study presented above); and (c) the fact that model-based projections themselves have the potential to influence policy decisions, e.g., optimistic model projections may lead to relaxed policies, which in turn will lead to worse outcomes than predicted, while pessimistic model projections may lead to stricter policies, which in turn will lead to better outcomes than predicted.

### 4.2 Future directions

More than a year after the emergence of SARS-CoV-2, our understanding of the pandemic is still evolving rapidly, especially regarding the risks posed by variant strains and the opportunities offered by vaccine candidates. These two issues currently present the most important questions regarding epidemic control, and hence are the two most active areas of Covasim development. Model parameter values are also continually updated as new data become available. Future development plans also include the incorporation of more detailed populations and networks, including healthcare workers, different types of industry, spatial mixing patterns, and the socioeconomic and racial disparities present in both transmission patterns and

health outcomes. With the deployment of vaccines comes additional questions and interest regarding the lifting of mobility restrictions and social distancing guidelines, as well as questions about equitable vaccine distribution to different populations around the world. Additional development of data-driven modeling of mobility between regions will help address the risk of importation to regions with fewer resources for early detection and treatment, as prepandemic mobility gradually returns to parts of the world. Finally, we are committed to continuing our collaborations with stakeholders and policymakers around the globe, to work with them in determining how COVID-19 suppression can be achieved via a combination of distancing, testing, contact tracing, and vaccination.

## Acknowledgments

Additional contributors to the Covasim model and this study include: from GitHub, William Fitzgerald, Hamel Husain, Cory Gwin, Julian Nadeau, Rasmus Wriedt Larsen, Aditya Sharad, and Oege de Moor; from Microsoft, William Chen, Scott Ayers, and Rolf Harms; from the Institute for Disease Modeling, Mary Fisher, Jennifer Schripsema, Dennis Chao, Christian Wiswell, Samuel Buxton, Christopher Lorton, Clinton Collins, Christopher Jones, Charles Eliot, Svetlana Titova, Dejan Lukacevic, Jeffrey Steinkraus, John Sheppard, Niket Thakkar, Roy Burstein, Robert Hart, Guillaume Chabot-Couture, Caitlin Bever, Helen Olsen, and Natalia Corona; from the Allen Institute, Natalia Orlova; from the Jet Propulsion Laboratory, Casey Handmer; from the QIMR Berghofer Medical Research Institute, Paula Sanz-Leon and James Roberts; from the Kirby Institute, Richard Gray; from the Burnet Institute, Tharindu Wickramaarachchi; from the University of California San Diego, Richard K. Belew; from the London School of Hygiene & Tropical Medicine, William Waites; and from Novosibirsk State University, Olga Krivorotko and Mariia Sosnovskaya. We also wish to thank the participants of the Covasim Users Group, including Julie Maher, Dean Sidelinger, and Erik Everson from the Oregon Health Authority; Samuel Mwalili and Duncan Gathungu from Jomo Kenyatta University of Agriculture and Technology; André Lin Ouédraogo from the Institute for Disease Modeling; David P. Wilson from the Bill and Melinda Gates Foundation; Edinah Mudimu, Brian Mudimu, and Chris Swanepoel from the University of South Africa; and Quang Duy Pham from the Pasteur Institute of Ho Chi Minh City.

## Author Contributions

**Conceptualization:** Cliff C. Kerr, Robyn M. Stuart, Dina Mistry, Romesh G. Abeysuriya, Michael Famulare, Daniel J. Klein.

**Data curation:** Cliff C. Kerr, Anna Palmer, Dominic Delport, Nick Scott, Sherrie L. Kelly, Caroline S. Bennette, Bradley G. Wagner, Stewart T. Chang, Assaf P. Oron, Daniel J. Klein.

**Formal analysis:** Cliff C. Kerr, Dina Mistry, Daniel J. Klein.

**Investigation:** Cliff C. Kerr, Dina Mistry, Rafael C. Núñez, Prashanth Selvaraj, Brittany Hagedorn, Jasmina Panovska-Griffiths, Daniel J. Klein.

**Methodology:** Cliff C. Kerr, Robyn M. Stuart, Dina Mistry, Romesh G. Abeysuriya, Katherine Rosenfeld, Gregory R. Hart, Rafael C. Núñez, Jamie A. Cohen, Prashanth Selvaraj, Brittany Hagedorn, Nick Scott, Michael Famulare, Daniel J. Klein.

**Project administration:** Cliff C. Kerr, Greer Fowler, Daniel J. Klein.

**Software:** Cliff C. Kerr, Robyn M. Stuart, Dina Mistry, Romesh G. Abeysuriya, Katherine Rosenfeld, Gregory R. Hart, Jamie A. Cohen, Lauren George, Michał Jastrzębski, Daniel J. Klein.

**Supervision:** Cliff C. Kerr, Nick Scott, Sherrie L. Kelly, Edward A. Wenger, Jasmina Panovska-Griffiths, Michael Famulare, Daniel J. Klein.

**Validation:** Cliff C. Kerr, Michael Famulare, Daniel J. Klein.

**Visualization:** Cliff C. Kerr, Robyn M. Stuart, Dina Mistry, Amanda S. Izzo.

**Writing – original draft:** Cliff C. Kerr, Robyn M. Stuart, Dina Mistry, Romesh G. Abeysuriya, Rafael C. Núñez, Brittany Hagedorn, Amanda S. Izzo, Daniel J. Klein.

**Writing – review & editing:** Cliff C. Kerr, Robyn M. Stuart, Dina Mistry, Romesh G. Abeysuriya, Katherine Rosenfeld, Rafael C. Núñez, Jamie A. Cohen, Amanda S. Izzo, Greer Fowler, Jasmina Panovska-Griffiths, Michael Famulare, Daniel J. Klein.

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
