## [Decision Letter · Decision Letter 0]

23 Mar 2021

Dear Dr Kerr,

Thank you very much for submitting your manuscript "Covasim: an agent-based model of COVID-19 dynamics and interventions" for consideration at PLOS Computational Biology.

As with all papers reviewed by the journal, your manuscript was reviewed by members of the editorial board and by several independent reviewers. In light of the reviews (below this email), we would like to invite the resubmission of a significantly-revised version that takes into account the reviewers' comments.

We cannot make any decision about publication until we have seen the revised manuscript and your response to the reviewers' comments. Your revised manuscript is also likely to be sent to reviewers for further evaluation.

Sincerely,

Manja Marz

Software Editor

PLOS Computational Biology

Manja Marz

Software Editor

PLOS Computational Biology

Reviewer's Responses to Questions

**Comments to the Authors:**

Reviewer #1: # Review Covasim

This paper introduces an agent-based simulation platform specialized on the COVID-19 pandemic. The organization, documentation and testing coverage of the code seems to have production quality, and in terms of performance and features reaches state-of-the-art. The paper is well written. We only have some very minors comments:

* In our view, the introduction would benefit if it would feature the most ''important'' points of the library, what set it appart from other agent-base models. What we personally found most interesting is:

* The dynamic rescaling feature, which seems to be a very elegant way to solve the computational problem of the interpolation between the low numbers stochastic nature of spreading and the mean-field dynamics dynamics at high case numbers.

* The possible integration with Optuna, a modern hyperparameter optimization library, which allows to calibrate the model by comparing its output to case and death numbers. This also allows a straightforward parallelizability of the simulations

* C-like performance because of the Numba integration

* It's success in predicting the decrease in infections because of the enactment of slight restrictions in the Seattle area.

* In the figure 6, the unit of the y-axis of the cpu-time plot is unclear. It doesn't seem to be cpu-seconds per simulation day as this would be a few orders of magnitude off from the performance example described in the text and the linear increase labelling in the plot.

* We had the impression, that in the current state, the model is best suited to simulate the propagation on a city level. How well could the model be extended to include another higher level network, for example the propagation between cities, states or countries? Some sort of mobility data is often available which allows a direct modelling of the network without making too much assumptions. If you find this point relevant it would eventually be interesting to touch on it in the discussion.

Viola Priesemann

Reviewer #2: This paper presents Covasim which is a comprehensive agent-based model for Covid-19. The paper describes the underlying model and software which is open-source and has been developed by a team from multiple institutions. It does not describe the details of calibration methods or results, which have been presented by the authors and external users of Covasim in other publications (i.e. not in scope for this paper). The model includes all the key aspects for modelling both the dynamics of the virus and disease, as well as interventions to reduce infections. The software is written to a high standard, transparent, and easy to use and extend.  A testament to this is that it has already been used by a number of external researchers beyond the core development team and has been used to advice policy makers in multiple countries. The paper is well written, clear and easy to read.

Specific comments and questions:

1. The default parameters correspond to a doubling time of 4-6 days and and an R0 of 2.2-27. Is the range stochastic uncertainty, if so what is the mean? Assuming the mean of these ranges, the doubling time seems a little high and the R0 seems a little low, certainly when compared to the early stages of the epidemic in Europe.

2. Fig 2 - Viral load timing. It seems that out of those who do go on to develop symptoms, their viral load (and thus infectiousness) will be zero prior to the onset of symptoms or non-zero for at most one day. What is the breakdown in transmission by symptom status of the source? Is the epidemic predominantly being driven by transmission from symptomatic individuals? What is the mean generation time? 

3. Fig 3 - not really necessary, all the information is in Fig 4 (which is very clear).

4. Contact tracing - what proportion of interactions are typically contact-traced? Does this depend upon the number of new infections (i.e. during very high incidence contract-tracing teams can get saturated)? How many days of prior interactions are traced? Have you modelled digital contract tracing, such as the Google-Apple Exposure Notification System?

5. Isolation/Quarantine - please can you give more details of how the probability of transmission is lowered. Is this done by changing the contact network (i.e. 80% of quarantined stay at home so only interact on their household network) or by a change in the per-interaction infectious rate? (or a mix of both)

6. Performance - considering the model is written in pure Python it has incredibly impressive performance, which is due to clever coding and the use of efficient packages such as Numba. What (if any) limitations does the array approach have compared to a more traditional object-oriented approach? Is performance dependent on the number of people infected and the interventions? If so, what are the conditions in the Fig 6. The text mentions that you use Sciris for parallelization, is the reported performance for a single processor and single thread, or are multiple processors and multi-threading being used?

7. Deployment - it is mentioned that it can be run in R using 'reticulate' (as can most Python code). Do you have a wrapper for Covasim in R using reticulate? Have you thought putting Covasim on CRAN or Bioconductor?

8. Software tests - please can you give more details of the tests (and examples).

9. Example usage - the example in Fig 9 is very impressive and demonstrates how simple/intuitive the code is for a complex simulation. Is it possible to have a dynamic intervention policy? Interventions such as lockdowns and school closure tend to be timed based on the prevailing incidence or hospital occupancy. Can this be modelled as opposed to interventions set between specific dates?

10. Case study - is a nice example which has been presented in a prior publication. What is the basis of the forecast interval? Is this the stochastic uncertainty of the model in a finite size population? Is it a Bayesian interval from uncertainty in the calibration of the model parameters?

**Have all data underlying the figures and results presented in the manuscript been provided?**

Reviewer #1: None

Reviewer #2: Yes

PLOS authors have the option to publish the peer review history of their article (what does this mean?). If published, this will include your full peer review and any attached files.

Reviewer #1: **Yes: **Viola Priesemann

Reviewer #2: No
---

## [Decision Letter · Decision Letter 1]

5 Jun 2021

Dear Dr Kerr,

We are pleased to inform you that your manuscript 'Covasim: an agent-based model of COVID-19 dynamics and interventions' has been provisionally accepted for publication in PLOS Computational Biology.

Best regards,

Manja Marz

Software Editor

PLOS Computational Biology

Manja Marz

Software Editor

PLOS Computational Biology

Reviewer's Responses to Questions

**Comments to the Authors:**

Reviewer #1: All points have been addressed.

This is a very nice paper!

Reviewer #2: The authors fully addressed all the issues I raised and adjusted the manuscript appropriately.

This model (and software) is now being used in the UK by the Joint Biosecurity Centre as one of the panel models used for weekly updating the UK government. As such, it an important tool in informing public health policy in the UK.

This is an excellent paper which I have no hesitation in recommending for publication.

**Have the authors made all data and (if applicable) computational code underlying the findings in their manuscript fully available?**

Reviewer #1: Yes

Reviewer #2: Yes

PLOS authors have the option to publish the peer review history of their article (what does this mean?). If published, this will include your full peer review and any attached files.

Reviewer #1: No

Reviewer #2: No

---

## [Editor Report · Acceptance letter]

21 Jul 2021

PCOMPBIOL-D-21-00272R1 

Covasim: an agent-based model of COVID-19 dynamics and interventions

Dear Dr Kerr,

I am pleased to inform you that your manuscript has been formally accepted for publication in PLOS Computational Biology. Your manuscript is now with our production department and you will be notified of the publication date in due course.

With kind regards,

Katalin Szabo
